# The Mechanical Properties of Reinforced Concrete Plate-Girders when Placed Under Repeated Simulated Vehicle Loads

**DOI:** 10.3390/ma12111831

**Published:** 2019-06-05

**Authors:** Jinquan Zhang, Pengfei Li, Yan Mao, Zhenhua Dong

**Affiliations:** Research Institute of Highway, Ministry of Transport of the People’s Republic of China, Beijing 100088, China; rioh_jqzhang@163.com (J.Z.); Y.mao@rioh.cn (Y.M.); dongzhenhua2009@126.com (Z.D.)

**Keywords:** bridge engineering, reinforced concrete plate-girders, repeated load, deflection, residual deflections

## Abstract

The effect of vehicle loads on reinforced concrete plate-girders was evaluated using the current Chinese specifications. Repeated loading performance tests with loading amplitudes of 77 kN, 97 kN, and 121 kN, which correspond to the standard vehicle load, 1.25 times overload, and 1.6 times overload proportions effect were carried out on three full-scale simply-supported reinforced concrete plate-girders. Our research results indicate that the development of cracks in reinforced concrete beams can be divided into three stages: rapid development, stability, and failure. During the entire process, the strain of steel and concrete did not reach their yield strain. The most severe damage done to the concrete beams was the brittle fractures caused by the fatigue fracturing of the rebar. When in a stable condition, the extent to which the vehicle was overloaded had a significant effect on the fatigue performance of the beam, and the corresponding residual deflection and residual strain increased with the rise in the overload proportion. In addition, as the overload proportion increased, the stiffness degradation and the cumulative damage that occurred under the same loading cycle was more significant. The test beam reached failure after being subjected to 350,000 and 670,000 repeated loading cycles, when the load was 1.6 times and 1.25 times of the standard load effect. With a standard vehicle load effect, the test beam was able to endure 2,000,000 repeated load cycles with no significant degradation in stiffness and bearing capacity.

## 1. Introduction

Concrete plate-girders with small or medium spans have been widely used in the construction of highway bridges due to their simple structure, low price, and the convenience of their fabrication. According to on-site inspections and the testing data obtained from existing bridges, damage gradually accumulates within concrete plate-girders, with strength degradation occurring due to the erosion caused by the external environment and the repeated effects of vehicle load. Moreover, the damage done to bridges which have endured overload for a long period of time tends to accumulate far more rapidly than the damage done to a bridge which has not been overloaded. Previous data has also verified that when the damage that has accumulated in the beam reaches a certain degree, sudden brittle failure of the bridge may occur, compromising the safety of the bridge and the surrounding traffic networks [1].

In order to analyze the damage caused by fatigue, a series of theoretical analyses and fatigue tests were carried out on ordinary reinforced concrete, high strength reinforced concrete, pre-stressed reinforced concrete, corroded reinforced concrete, and different forms of reinforced concrete subjected to a range of environmental impact factors. The fatigue development law of the reinforced concrete members [2], the influence of the strength of reinforced concrete materials [3], the stress amplitude of rebar, and the corrosion rates of rebar on the fatigue performance and fatigue life of concrete beam elements were all studied based on the analysis of the variation trends of the deflection, strain, and bearing capacity of reinforced concrete members when placed under the fatigue loads [4,5,6]. Damaged reinforced concrete beams strengthened with CFRP (Carbon Fiber Reinforced Plastics), AFRP (Aramid Fiber Reinforced Plastic), steel plates, and other methods were also studied in order to analyze the effects of different reinforcement materials and methods on the fatigue behavior of reinforced concrete members [7,8,9,10,11,12,13]. In addition, the fatigue damage model, damage development law, fatigue life design method, and the residual life evaluation of reinforced concrete beams were also studied and different stress amplitude–number of cycles (S–N) curve models of the reinforced concrete members were obtained based on the results of existing research [14,15,16,17]. However, in conventional fatigue test methods for reinforced concrete members, the amplitude of the cyclic loadings is determined according to a certain proportion of the design values or the measured values of the ultimate bearing capacity of the test specimens, and this amplitude can not accurately reflect the load effects on the structure under actual, real-world conditions. Meanwhile, the phenomenon of overloaded transportation is a serious issue in some areas of China, which has resulted in bridges experiencing a series of cyclic loadings larger than their design values. The fatigue or cumulative damage caused by this problem was more prominent in small and medium span bridges due to the large proportion of the vehicle live loads in the total load effect. 

The purpose of this study is to analyze the mechanical behavior of small and medium span concrete bridges under a series of cyclic vehicle loads in different overload proportions. Three full-scale prototype concrete plate-girders were designed. A series of cyclic loading tests were conducted with loading amplitudes of 77 kN, 97 kN, and 121 kN, which correspond to the standard vehicle load, 1.25 times overload, and 1.6 times overload proportion effect. Through the tests, we analyze the failure law, deflection and strain variation law, cumulative damage evolution law, and the stiffness degradation law of the concrete plate-girders in different overload proportions, and the relationship between fatigue life of reinforced concrete plate-girders to overload proportions is fitted.

## 2. Test of the Specimens

### 2.1. Design of the Beams

Three full-scale prototype concrete plate-girders with a length of 6 m were designed according to the general principles presented in Highway Bridge and Culvert Standard Drawings (1973–1993), named as Beams-1–3. The dimensions of the test specimens are shown in Figure 1. C50 concrete material was used in the design of the beam members and the average experimental compressive strength of standard concrete cubes was 56.8 MPa. Seven lengths of HRB400 rebar with an ideal yield strength of 400 MPa and a diameter of 25 mm were set evenly along the bottom of the concrete plate-girders. The measured tensile yield strength and the ultimate tensile strength of the steel rebar were 436 MPa and 634 MPa, respectively. HRB335 round rebar with an ideal yield strength of 335 MPa and a diameter of 8 mm was used for the stirrups, and a spacing of 100/200 mm was employed. The measured tensile yield strength and the ultimate tensile strength of the stirrups were 359 MPa and 527 MPa, respectively. Three HRB335 longitudinal bars with a diameter of 8 mm were placed at the top of the beams, and the thickness of the protection layer was 45 mm. In order to simulate the effect of actual vehicle load on the beams and determine the loading scheme of the tests, a two lane bridge with a width of 8 m and a span of 6 m was designed using the concrete plate-girders, and the calculated span $l_{0}$ was equal to 5.7 m, as shown in Figure 2.

Transverse joints were used for the beam connections, and the stiffness parameter *γ* was determined as
(1)$$\gamma =5.8\frac{I}{I_{t}}{\left(\frac{b}{l}\right)}^{2}=0.0524$$
where the parameters *I* and *I*_t_ were the bending moment of inertia and torsion moment of inertia of the beams; *b* and *l* were the width and height of the rectangular beam section.

The influence curve of the transverse distribution coefficient of the 1#–4# beam specimens is shown in Figure 3.

According to provisions stated in the General Specifications for the Design of Highway Bridges and Culverts (JTG D60-2015), the transverse and longitudinal arrangements of highway grade I vehicle load is shown in Figure 4. The maximum load affecting the middle span of each beam under the action of rear axles was calculated on the basis of both of the influence curve of the transverse distribution of beams and the transverse arrangement of vehicle loads. Assuming that the transverse connections between the beams are intact, and the vehicle loads are arranged along the outermost sides. In this case, the maximum equivalent load acting on Beam-2 is 77 kN. In order to simulate the effects of actual vehicle loads on bridges, the amplitudes of cyclic loadings were set as 77 kN, 97 kN, and 121 kN in order to assess the performances of the beams at 1.0, 1.25, and 1.6 times of the highway grade I vehicle load, as listed in Table 1.

### 2.2. Loading scheme

In order to achieve a pure curved section in the middle portion of the beam specimens, and to simulate the actual axle action of a vehicle equivalent to a highway grade I vehicle load, the forces were loaded using distributive beams with a spacing of 1.4 m. Ordinary plate rubber bearings were set at both ends of the test specimens, as shown in Figure 5. A series of vertical cyclic loading tests was conducted using a hydraulic jack.

A pre-loading not exceeding 70% of the normal service load was applied to the specimens to ensure that the instruments were functioning correctly. Monotonic static loading and unloading behaviors were conducted before the cyclic loading tests, and the process of both behaviors were divided into four stages. The cyclic loading tests were conducted at a frequency of 4 Hz. The residual deflection, the distribution, and the width of the cracks appearing on the beam specimens were measured after every 100,000 load cycles. Then a monotonic static loading of four stages was repeated to measure the strain on both the concrete and rebar, the crack width, and the deflection of the beam specimens at each stage. The distribution of cracks and the failure characteristics of the specimens were also recorded.

### 2.3. Measurement scheme

In order to measure deflection at different positions along the beam span, displacement meters were installed at the middle span, the loading point, at the one-sixth span position, and at both ends of the beams. Additionally, concrete strain gauges were arranged along the perimeter of the beam sections, at both the middle span and the loading point of the test beam, in order to measure the concrete strain. Strain gauges were also arranged on seven of the lengths of longitudinal rebar in order to measure the strain experienced by the rebar in different positions, as shown in Figure 6.

## 3. Analysis of the Test Results

### 3.1. Test Phenomena

Under cyclic loading, brittle fractures occurred along main cracks near Beam-2 and Beam-3 but there was no obvious indication that failure was about to occur. During the failure process of Beam-2 and Beam-3, the amount of cracks and their width and height along the length of the beams experienced three stages: the early rapid development stage, the middle stability stage, and the final failure stage. Initially, when the loading of Beam-2 had reached the 100,000th cycle, all of the cracks in the specimen were half the height of the beam height. The cracks were distributed symmetrically along the span direction and on both sides of the cross section. Both the number of cracks and their width remained constant when the number of loading cycles increased from 100,000 times to 600,000 times, with the beam continuing to remain in the crack stability stage. Finally, a dominant crack in the beam span suddenly extended and widened when the 650,147th cycle was achieved, resulting in a fracture in the beam specimen. At this point, the beam underwent a sudden, major deformation and lost its bearing capability due to the ruptures of the five rebar steels in the bottom of the beam. The concrete in the compression zone was raised, accompanied by shedding. 

Similar behaviors occurred when Beam-3 reached the 10,000th load cycle but in this case the height of the cracks that developed along the side faces were larger. Beam-3 reached the crack stability stage between loadings of 100,000 times to 300,000 times. However, when the load cycles reached 377,240 an abrupt fracture occurred due to the rupture of the rebar in the bottom of the specimen. This rupture occurred without obvious precursors. Easily identifiable fatigue failure characteristics can usually be found on the fracture surface of the rebar, but in this case no necking phenomenon was observed. The limit failure modes of Beam-2 and Beam-3 are shown in Figure 7. Excluding the main cracks, there was little variation in the width of the additional cracks in Beam-2 and Beam-3. Before the overall fracture of the beam specimens occurred, the maximum residual crack width was about 0.08 mm.

No fractures occurred in Beam-1 after 2,000,000 loading cycles with a maximum loading force of 77 kN. At the base of the beam, the penetrating cracks were distributed uniformly along the length of the beam. The cracks also extended up to half of the height of the web, with a spacing of 20 cm. The cracks tended to close up following the unloading stage, with a maximum crack width of 0.1 mm. When adjacent to the middle part of the beam, residual crack widths were generally less than 0.08 mm. Throughout the cyclic loading process, the crack development of Beam-1 could be divided into the rapid development stage, the slow development stage, and the stable stage. The final condition of Beam-1 and its crack distribution is shown in Figure 8. In this instance, all of the cracks occurred when the load cycle had been performed 100,000 times. Beam-1 displayed more cracks than Beam-2 and Beam-3, with most of the cracks located 10 cm below the beam height. A new crack occurred at the end of the beam, on both sides, when the load cycling had been performed 1,000,000 times. The crack developed gradually, spreading up the beam and reaching about half the overall height of the beam. The number, height, and width of the cracks did not increase when the number of loadings increased from 1,000,000 times to 2,000,000 times.

A destructive test was conducted on Beam-1 after it had been subjected to 2,000,000 cyclic loadings. Finally, an abrupt fracture occurred in the Beam-1 along the cracks in the middle span with a bearing force *P* of 268 kN, which was defined as the residual bearing capacity of the test beam. This demonstrates that the difference between the theoretically calculated value of the ultimate bearing capacity of the beam and the residual bearing capacity was very small. Additionally, according to the test phenomena, a linear relationship exists between the force and the deflection of the mid span before the force exceeded 260 kN, indicating that the beam was basically in an elastic stage. A displacement-control loading scheme was used when the force was loaded to 260 kN, and the bearing capacity of the beam was able to remain stable for a long time during the increase in deflection that occurred when the ultimate load was reached. The ductile failure of the beam, with an ultimate deflection in the mid-span of 58 mm, was caused by the tensile yield of the bottom rebar. The residual bearing capacity of the beam was more than 90% of the theoretical ultimate bearing capacity, indicating that the ultimate bearing capacity of the reinforced concrete beams was largely provided by the tensile strength of the rebar. Under 2,000,000 standard load cycles, there was no obvious decrease in the bearing capacity of reinforced concrete beams, indicating that the residual bearing capacity of Beam-1 meets bearing capacity design requirements. However, the service life of Beam-2 and Beam-3 decreased rapidly when subjected to the overloaded cycles.

### 3.2. Analysis of Deflection and Stiffness Degradation

The deflection curves of the different measuring points along the span of Beam-1, Beam-2, and Beam-3 under different cyclic loads of varying amplitudes are shown in Figure 9. For the test of Beam-1, the maximum mid-span deflection was 9.56 mm after 2,000,000 loading cycles, and no obvious change in vertical deflection was detected along the span as the loading cycles increased. The maximum mid-span deflection was only 1.5% higher than the deflection caused by the initial static load, indicating that the beam was basically in an elastic stage. For Beam-2, the vertical deflection that occurred at different positions along the beam span under peak static load was significantly higher than the deflection that occurred under the initial static load. For Beam-2, the mid-span deflection increased by 13.7% after 10,000 loading cycles. From this point the deflection increased slowly, and vertical deflection increased by 20.4% compared with the initial static load after 300,000 cyclic loading cycles. A brittle fracture failure occurred in the mid-span of Beam-2 when the loading cycle reached 650,000. When Beam-2 had experienced 600,000 loading cycles, the mid-span deflection under the peak load was 14.3 mm, 24.5% higher than the deflection that occurred under the initial static load. The deflection–span ratio was about 1/420.

Beam-3 displayed similar variation in vertical deflection to that of Beam-2. The mid-span deflection under peak loading was 13.6% higher than the deflection under the initial static load when the loading cycle had been repeated 10,000 times. The vertical deflection of the mid-span increased suddenly and obviously after 50,000 cyclic loadings, and the maximum deflection of the mid-span increased by about 23.8% compared with the initial value. The mid-span deflection was 19.75 mm, which is 30.8% higher than the initial static load after 300,000 loading cycles, and the deflection–span ratio of the beam was about 1/300. Finally, a brittle fracture failure occurred in the span of Beam-3 after to 370,000 loading cycles. During the entire cyclic loading process, the deflection of Beam-2 and Beam-3 increased nonlinearly with the increase in loading cycles, indicating that the damage to the beam accumulated gradually under repeated vehicle loads, while the stiffness of the members lessened.

The vertical deflection of the different beams’ mid-span under a range of loading cycles is shown in Figure 10. When the overload effect is not considered, Beam-1 was able to survive 2,000,000 cyclic loads. The deflection of Beam-1 was essentially stable as the cycles increased, and there was no serious or irrecoverable deformation. For the destructive tests of Beam-2 and Beam-3, the vertical deflection of each section underwent the process of increasing–stabilizing–brittle failure with the increase in cycles. The deflection stability stage was the same as the fatigue damage process of the tensile rebar at the bottom of the beams. In addition, the deflection development of Beam-3 was significantly faster than that of Beam-2, indicating that a more serious cumulative damage can be achieved by increasing the amplitude of the cyclic loadings. Therefore, the repeated actions of overloaded vehicles can increase the deformation of concrete beams, and accelerate the development of internal damage. The higher the proportion of overload is, the more obvious this trend.

Bending stiffness is one of the most important indicators used characterize the performance of reinforced concrete beams. The deflection analysis shows that the stiffness of the test beams decreased gradually with the increase of load cycles. In the test, the bending stiffness of the beam can be reflected by the load–deflection ratio of the mid-span section under different loading cycles, as shown in Figure 11. For the test of Beam-1, the load–deflection ratio curve decreases slowly, indicating that the bending stiffness of the beam decreased slightly with the increasing cyclic loading, and the bending stiffness of the beam decreased by 4.4% compared with the maximum value after 2,000,000 cycles. For Beam-2, the load–deflection ratio decreased rapidly after 10,000 cycles, and then decreased gradually until the final failure stage, indicating that the bending stiffness of the beam decreased gradually with the increase in loading cycles; after 60,000 cycles the stiffness was 48.7% lower than the initial stiffness. For the test of Beam-3, the load–deflection ratio maintained a relatively fast descent rate during 50,000 cycles, and the descent rate slightly decreased during 50,000 to 300,000 cycles but was still higher than that of Beam-2. The ultimate bending stiffness of the Beam-3 following 300,000 cycles was 55.4% lower than its bending stiffness under the static load.

Our results indicate that the bending stiffness of the beam did not experience a gradual, stable degradation process under the cyclic vehicle loads; its change trend was related to the values of vehicle loads. Under the repeated action of a standard load, the stiffness of the beams did not degenerate significantly, and the members remained in an elastic stage. However, under 1.6 times the standard load, the bending stiffness of the beam decreased significantly with an increase of loading cycles, indicating that the beam was irreversibly damaged. Therefore, the ratio of overloaded vehicles and the number of passages should be strictly controlled in order to improve the mechanical performance of beams and extend the service life of the overall structure. 

### 3.3. Analysis of the Residual Deflection

The development of the internal damage done to the beams directly corresponds with residual deflection. The residual deflection curves along the key sections of Beam-1, Beam-2, and Beam-3 after different loading cycles are shown in Figure 12. For the test of Beam-1, under a vertical load amplitude of 77 kN, the residual deflection increased linearly and slowly along the different positions of the span with the increase in the number of loading cycles, and the residual deflection decreased gradually from the mid-span to both ends. At 2,000,000 loading cycles, the maximum residual deflection of section D-4 was 1.8 mm, and the maximum residual deflection was 19.5% of the absolute deflection. For the test of Beam-2, under a vertical load amplitude of 97 kN, the vertical residual deflection along the different positions of the span increased significantly with the increase in loading cycles. After 300,000 loading cycles, the maximum residual deflection of the mid span section was 3.0 mm, which accounted for 21.6% of the absolute vertical deflection. After 600,000 loading cycles the maximum residual deflection of the mid span section was 4.8 mm, which accounted for 30.75% of the absolute vertical deflection. For the test of Beam-3 with a vertical load amplitude of 121 kN, the variation trend of the vertical residual deflection was similar to that of Beam-2. The residual deflection increased rapidly with the increase in the number of loading cycles. After 300,000 loading cycles, the maximum residual deflection in the middle span was 3.3 mm, which accounts for 18.5% of the absolute vertical deflection.

The comparison of mid-span maximum residual deflection for the different test beams under a range of loading cycles is shown in Figure 13. Our results indicate that residual deflection increased with the increase of the vertical load amplitude over the same amount of cycles, and that the overload ratio influences the beam’s degree of cumulative damage.

According to our analysis of deflection and residual deflection, the absolute vertical deflection of each section along the span of Beam-1 changed slightly with the increase in cyclic loading times, and the corresponding residual deflection also experienced a slow increase. The deflection of Beam-1 was basically stable after 1,000,000 cycles, and the stiffness of the beam ceased to degenerate. For Beam-2 and Beam-3 the vertical absolute deflection of different sections increased with the increase in cyclic loading times, and the residual deflection also experienced a significant increase. The residual deformation of Beam-3 was notably larger than that of Beam-2 after experiencing the same number of loading cycles. This indicates that the plastic cumulative damage done to the beams increases gradually with an increase in repeated vehicle loads.

### 3.4. Strain and Stress Analysis

During the cyclic loading process, the average strain on the rebar and concrete in a section of the beam was taken to be representative of the corresponding strain in that section. The variation trend displayed by the average longitudinal reinforcement tensile strain and the average concrete compressive strain along the different sections of the beam span are shown in Figure 14 and Figure 15. The maximum tensile strain of the longitudinal reinforcement in Beam-1 was 855.1 με, and the strain on the rebar did not alter significantly with an increase of loading cycles. The rebar in Beam-1 did not reach yield strain throughout the entire testing process. The maximum tensile strain of the longitudinal reinforcement in Beam-2 was 1765.3 με, which underwent several stages of development, including sudden increase–stable–accelerated growth–stable strain development with the increase of loading cycles. The maximum tensile strain of the longitudinal reinforcement at mid-span in Beam-3 was 1858.7 με, which underwent sudden strain increase and stable strain development with the increase of loading cycles.

According to our observations, the sudden increase in the strain on the bottom tensile rebar corresponds with the generation and rapid development of the cracks in the concrete beam. The stress is gradually transferred to the rebar as the concrete slowly develops cracks. The rebar was forced to bear the cyclic loads when the cracks developed completely. At this stage, the strain on the rebar entered the stable development stage under static load with a stable peak load. The strain on the rebar underwent a smaller increase under the same load because in this case the damage to the rebar had not yet developed. Until the fatigue fracture of the rebar occurred, the maximum strain in the rebar did not reach yield strain.

Our results indicate that during the cyclic loading, when the simulated vehicle loads were 1.6 times larger than the standard load, the rebar was constantly in a high-stress state. This resulted in a rapid increase in fatigue and cumulative damage to the plasticity of the rebar. The ultimate failure of the beam was caused by the rebar’s development of a fatigue crack, not due to the yield failure of the rebar itself.

Figure 15 shows the variation in the maximum compressive strain on the concrete at the compressive edge of the top of the test beam. The maximum strain was the average strain at the compressive edge of the concrete when the static load reached its maximum value after a specific number of cycles.

During the initial loading stage, when the number of cycles was below 50,000, the concrete’s strain grows rapidly. As the cracks develop it gradually enters the stable stage. In the second stage, when the number of cycles was between 50,000 and 90% of the total number of loading cycles, the compressive strain growth of the concrete was small and the cracks also entered the stable development stage. In the third stage, when the number of cycles was more than 90% of the total number of loading cycles, the compressive strain of the concrete entered an ascending stage, ultimately leading to the failure of the specimens. The strain of the concrete at the compressive edge of the top of the beam did not reach yield strain during the entire loading process. The mechanical behavior of the concrete beams under cyclic loading is largely determined by the presence of tensile steel bars. 

According to our strain analysis of the rebar and concrete, the compressive strain of concrete and the tensile strain of the rebar fluctuated slightly within a certain range but generally the strain curves developed smoothly. The maximum tensile strain on the rebar and the maximum compressive strain on the concrete did not reach their ultimate strain values. Even though fractures occurred during the testing of Beam-2 and Beam-3, the rebar still did not reach yielding strain. Our results indicate that the failure of the beams was caused by the fatigue fracture of the tensile rebar, which is in accordance with the fatigue failure characteristics of reinforced concrete beams. 

In order to obtain the law of material strain distribution in the critical stress section, the top concrete compressive strain, web concrete compressive strain, tensile strain, and bottom longitudinal reinforcement tensile strain in the same section were measured, and their distribution along the section height of the beams is shown in Figure 16, Figure 17 and Figure 18.

It can be observed that the strain distribution of the concrete across different sections of the test beams possessed linear characteristics, which conforms to the plane section assumption. After 10,000 loading cycles the position of the neutral axis increased compared with the initial static load, and the position of the neutral axis was seen to increase slowly as the number of cycles rose. Because Beam-1 was in an elastic state during the loading process, the rebar’s strain under different cyclic cycles remained stable. However, due to a gradual accumulation of residual strain in the rebar during the testing process, the performance of Beam-2 and Beam-3 were gradually degraded, and the strain on the rebar increased gradually during different loading cycles. Our results indicate that the fatigue damage experienced by the steel bars was small and the functional performance of the beams remained stable under the repeated action of standard vehicle loads. However, when subjected to the repeated action of overloaded vehicles the damage to the rebar accumulated quickly and their performances degenerated significantly, leading to the early fatigue failure of the beams. 

Based on our analysis of the fatigue test results for reinforced concrete beams, data with similar conditions were selected. The fatigue life data of different specimens under different stress levels as stated in reference [3,18,19,20] are shown in Figure 19. The stress amplitude–number of cycles (S–N) curve of ordinary reinforced concrete beams can be obtained by fitting these data, and the stress amplitude (*S*) can be calculated as,
*S* = 0.006*N*^2^ − 2.32*N* + 407.88(2)
where *N* is the number of cycles. Therefore, the relationship between the vehicle loads and the number of cycles can also be deduced from the results of this study. For example, when the vehicle load was 1.25–1.6 times the standard load, the number of cyclic loads experienced by the concrete beams was between 200,000–750,000.

## 4. Conclusions

This paper investigated the mechanical performance of reinforced concrete plate-girders when subject to long-term cyclic vehicle loading. Our results indicate that the failure process of reinforced concrete beams under cyclic loading can be divided into three stages: the rapid development stage, the stable stage, and the failure stage. The mechanical behavior of reinforced concrete plate-girders under cyclic loading proved to be largely determined by the rebar. Although many cracks appeared in the concrete Beam-1 that was subjected to the standard loads, the overall bearing capacity and stiffness of the beams under cyclic loading showed almost no obvious degradation due to the small amount of accumulated damage in the rebar. With an increase in the overload ratio, the mid-span deflection and residual deflection increased gradually, and the bending stiffness of the beam decreased significantly, while the cumulative damage increased gradually. During the entire cyclic loading process, the tensile strain of the rebar and the compressive strain of the beams’ concrete was less than their ultimate strain, and the ultimate failure of the beam was caused by the sudden failure generated by the fatigue fracture of the tensile reinforcing bar. The overload proportion had a strong influence on the failure state and performance degradation law of the beams. The beams were not damaged after 2,000,000 cycles of a standard vehicle load, whereas fatigue fracture occurred when the vehicles were overloaded. The cyclic loadings with overload ratios of 25% and 60% had cyclic times of 670,000 and 350,000, respectively. Therefore, the overload ratio of vehicle loads should be strictly controlled in order to avoid or reduce the possibility of highway bridges experiencing structural brittle failure.

## Figures and Tables

**Figure 1 materials-12-01831-f001:**
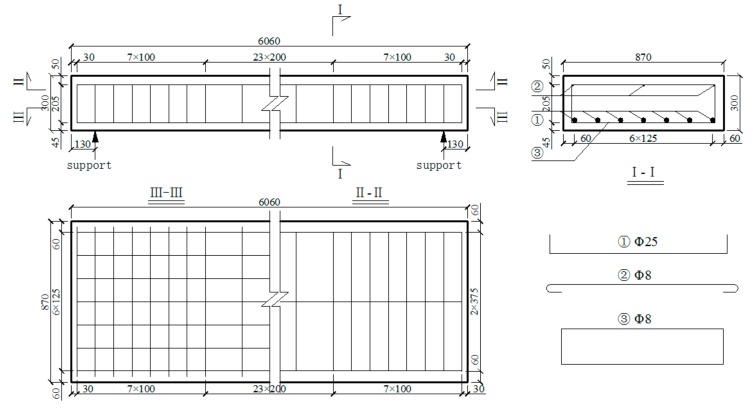
Specimen size and reinforcement drawing (mm).

**Figure 2 materials-12-01831-f002:**
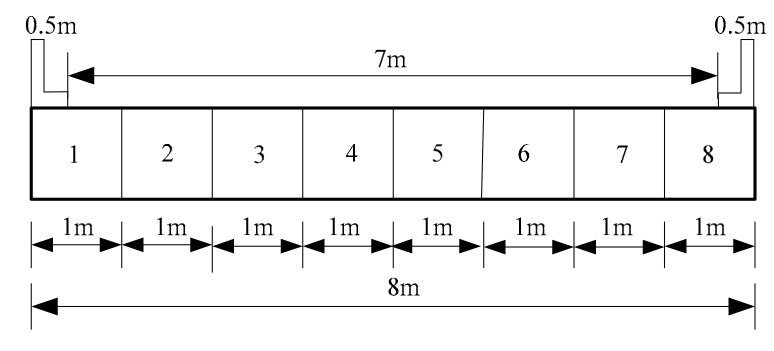
Bridge cross section layout.

**Figure 3 materials-12-01831-f003:**
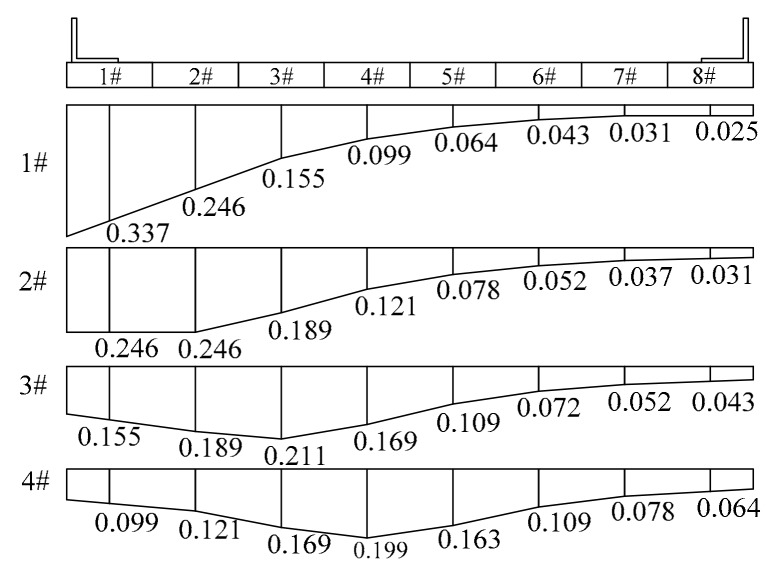
Horizontal distribution coefficient influence line.

**Figure 4 materials-12-01831-f004:**
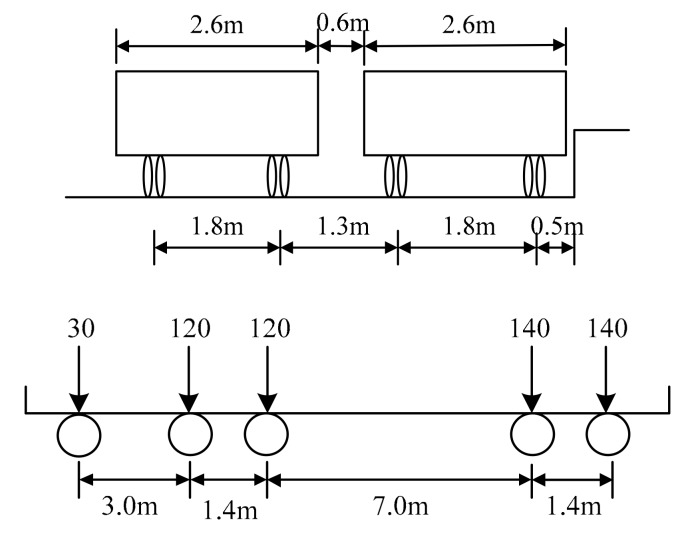
The vehicle load arrangement (kN).

**Figure 5 materials-12-01831-f005:**
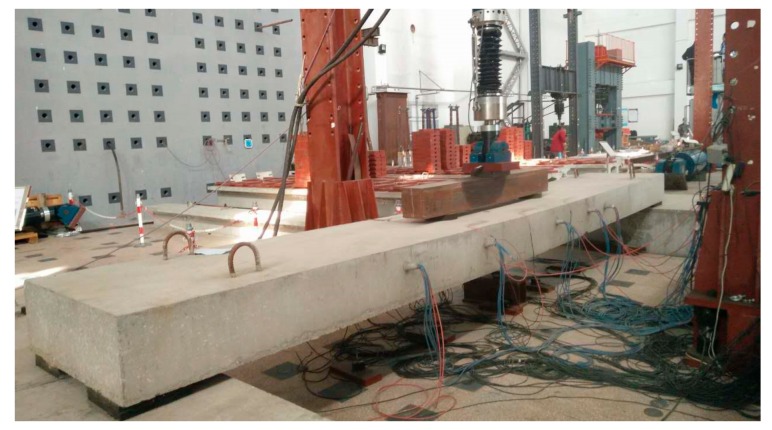
Diagram of loading device.

**Figure 6 materials-12-01831-f006:**
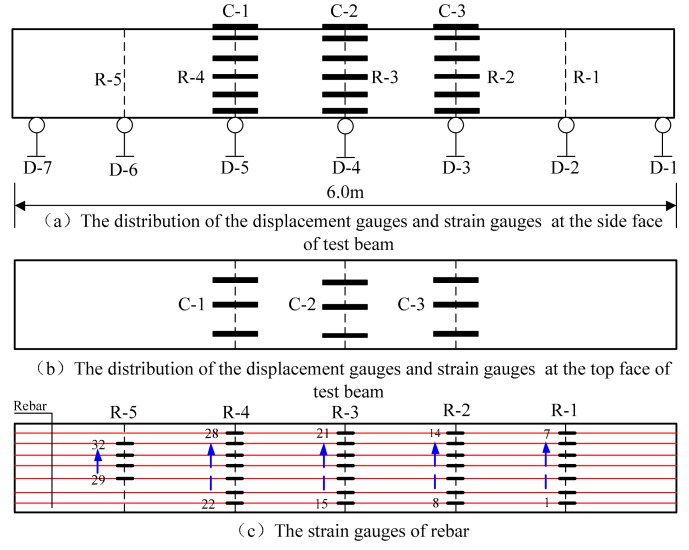
Diagram of the measuring scheme and measuring point layout.

**Figure 7 materials-12-01831-f007:**
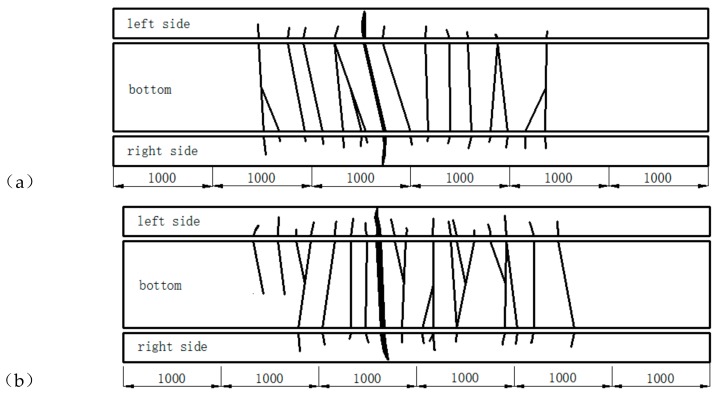
The collapse state of (**a**) Beam-2 and (**b**) Beam-3 (mm).

**Figure 8 materials-12-01831-f008:**
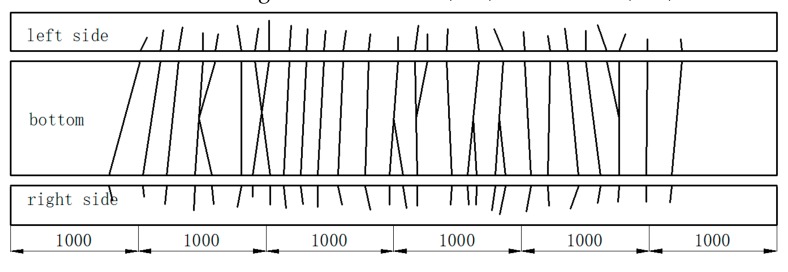
Beam-1′s crack distribution (mm).

**Figure 9 materials-12-01831-f009:**
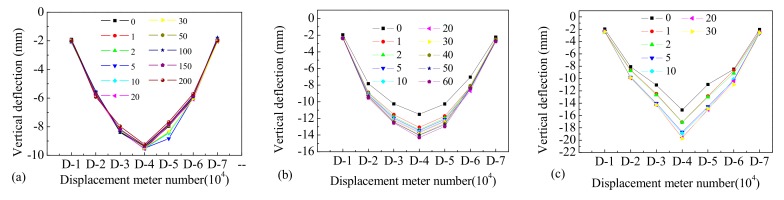
The deflection changes of (**a**) Beam-1, (**b**) Beam-2, and (**c**) Beam-3.

**Figure 10 materials-12-01831-f010:**
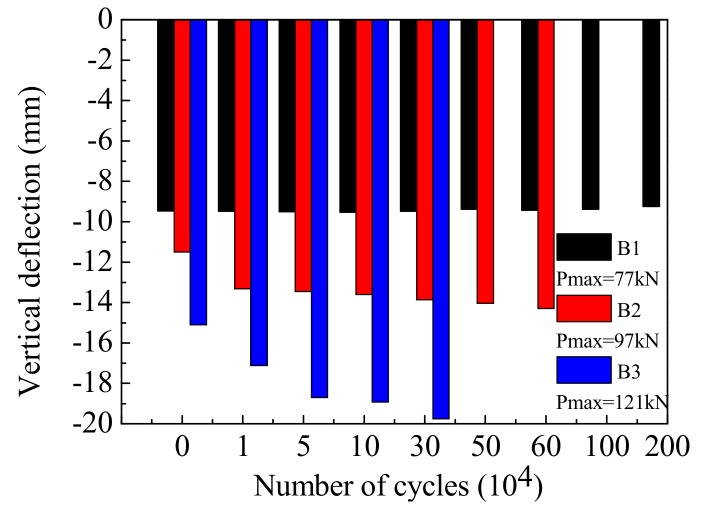
A comparison of maximum mid-span deflection across different cycle times.

**Figure 11 materials-12-01831-f011:**
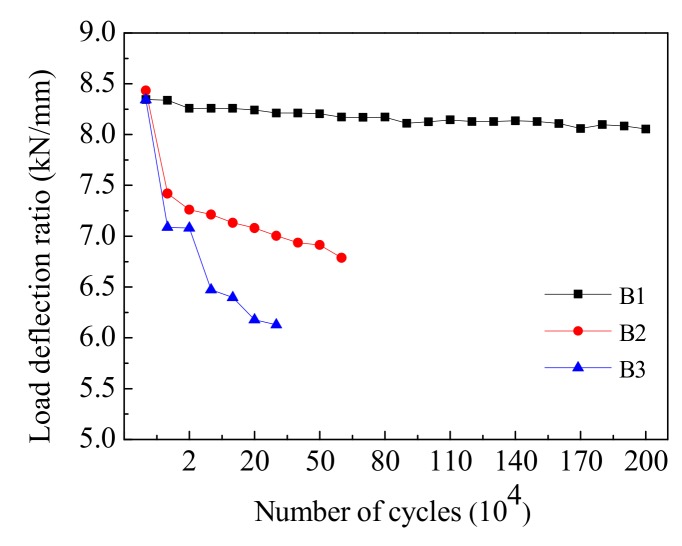
The load deflection ratio of test beams.

**Figure 12 materials-12-01831-f012:**
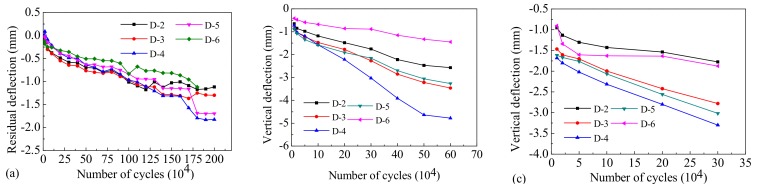
Residual deflection trend of (**a**) Beam-1, (**b**) Beam-2, and (**c**) Beam-3.

**Figure 13 materials-12-01831-f013:**
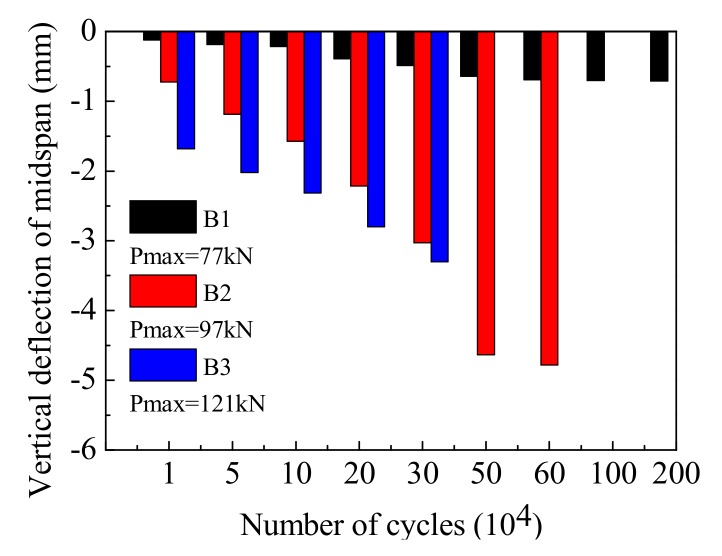
A comparison of the maximum mid-span residual deflection across different cycle times.

**Figure 14 materials-12-01831-f014:**
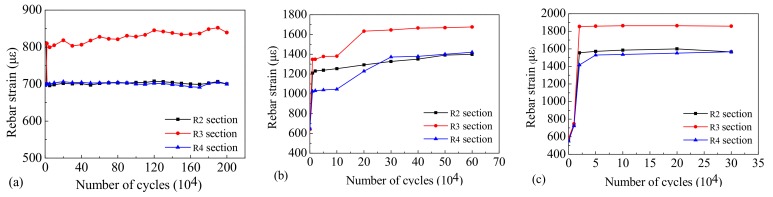
The strain variation trend of the rebar in (**a**) Beam-1, (**b**) Beam-2, and (**c**) Beam-3.

**Figure 15 materials-12-01831-f015:**
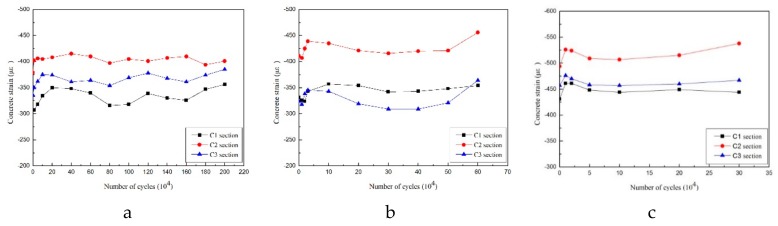
Variation trend of compressive strain on concrete of (**a**) Beam-1, (**b**) Beam-2, and (**c**) Beam-3.

**Figure 16 materials-12-01831-f016:**
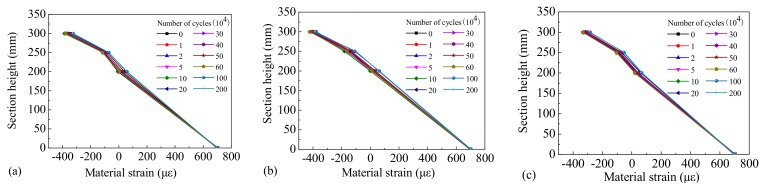
Strain distribution of Beam-1 of (**a**) section C-1, (**b**) section C-2, and (**c**) section C-3.

**Figure 17 materials-12-01831-f017:**
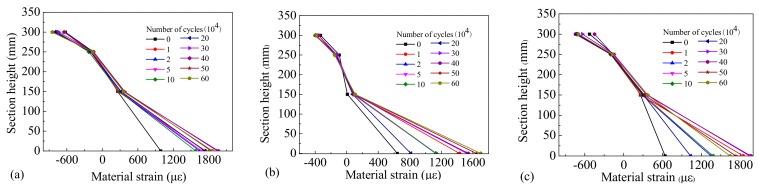
Strain distribution of Beam-2 of (**a**) section C-1, (**b**) section C-2, and (**c**) section C-3.

**Figure 18 materials-12-01831-f018:**
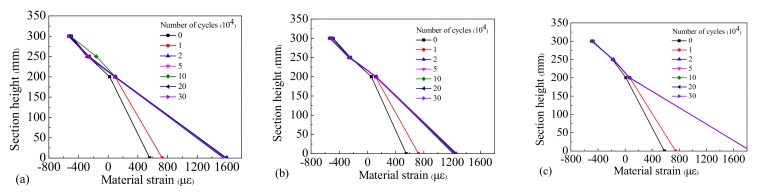
Strain distribution of Beam-3 of (**a**) section C-1, (**b**) section C-2, and (**c**) section C-3.

**Figure 19 materials-12-01831-f019:**
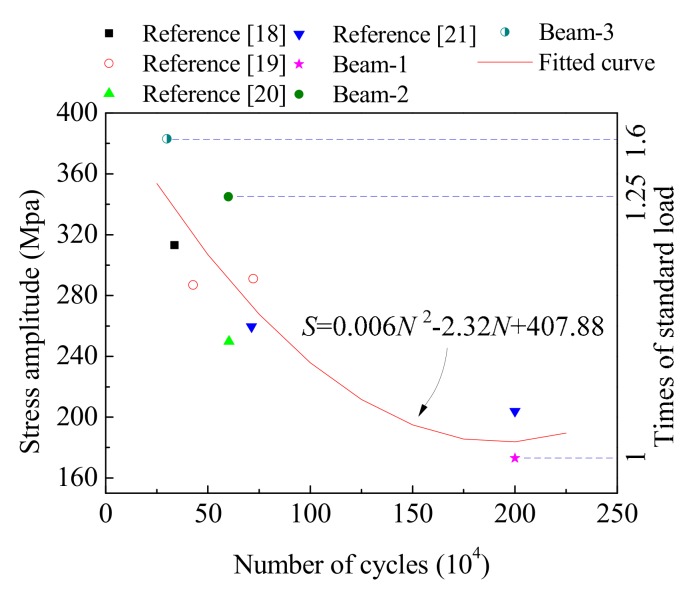
Fitting life-overload proportion curve of reinforced concrete beams.

**Table 1 materials-12-01831-t001:** Model test condition settings.

Specimen No.	Maximum Load (kN)	Ratio to Standard Vehicle Load
Beam-1	77	1.0
Beam-2	97	1.25
Beam-3	121	1.6

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
