# Peer review of "The Mechanical Properties of Reinforced Concrete Plate-Girders when Placed Under Repeated Simulated Vehicle Loads"

_materials, 2019, doi:10.3390/ma12111831_

Round 1
Reviewer 1 Report
The manuscript presents the results of an experimental study investigating the behaviour of slabs under high cycle fatigue. While generally interesting and well explained (although requiring some English language editing) the manuscript does require some improvements before publication.
It is not clear what the original contribution of the manuscript is, many other experimental studies of high cycle fatigue of flexural members exist, what is the significance of this one?
The literature review needs significant improvement to place this work in the context of what has already been done.
What are slab beams? is this a technical term in the Chinese standards as it does not appear to be common technical language and hence is confusing without further description.
Author Response
1.It is not clear what the original contribution of the manuscript is, many other experimental studies of high cycle fatigue of flexural members exist, what is the significance of this one?
Reply: Although a series of experimental studies of high cycle fatigue of flexural members have been conducted, the loading amplitude of conventional tests is determined according to a certain proportions of the design values or the measured values of the ultimate bearing capacity of the test specimens, which can not accurately reflect the load effects on the structure under actual, real-world conditions. Meanwhile, the phenomenon of overloaded transportation is a serious issue in some areas of China, which has resulted in bridges experiencing a series of cyclic loadings larger than their design values. The fatigue or cumulative damage caused by this problem was more prominent in small and medium span bridges due to the large proportion of the vehicle live loads in the total load effect.
Therefore, the purpose of this manuscript is to study the mechanical behavior of small and medium span concrete bridges under a series of cyclic vehicle loads in different overload proportions. For this purpose, three full-scale prototype concrete plate-girders were designed. A series of cyclic loading tests were conducted, with loading amplitudes of 77kN, 97kN, and 121kN, which corresponding to the standard vehicle load, 1.25 times overload and 1.6 times overload proportions effect. Through the tests, we analyze the failure law, deflection and strain variation law, cumulative damage evolution law and the stiffness degradation law of the concrete plate-girders in different overload proportions, and the curve of fatigue life of reinforced concrete plate-girders with overload proportions is fitted.
2. The literature review needs significant improvement to place this work in the context of what has already been done.
Reply: The literature review in the manuscript had been modified according to the opinions of reviewer 2, please see our revisions on p. 1-2.
“The fatigue development law of the reinforced concrete members[2], the influence of the strength of reinforced concrete materials[3], the stress amplitude of rebar and the corrosion rates of rebar on the fatigue performance and fatigue life of concrete beam elements were all studied based on the analysis of the variation trend of the deflection, strain and bearing capacity of reinforced concrete members when placed under the fatigue loadings[4-6]. Damaged RC beams strengthened with CFRP, AFRP, steel plates and other methods were also studied in order to analyze the effects of different reinforcement materials and methods on the fatigue behavior of reinforced concrete members[7-13]. In addition, the fatigue damage model, damage development law, fatigue life design method and the residual life evaluation of reinforced concrete beams were also studied and different S-N curve model of the reinforced concrete members were obtained based on the results of existing research[14-17]. However, in conventional fatigue test methods for reinforced concrete members, the amplitude of the cyclic loadings is determined according to a certain proportions of the design values or the measured values of the ultimate bearing capacity of the test specimens, and this amplitude can not accurately reflect the load effects on the structure under actual, real-world conditions. Meanwhile, the phenomenon of overloaded transportation is a serious issue in some areas of China, which has resulted in bridges experiencing a series of cyclic loadings larger than their design values. The fatigue or cumulative damage caused by this problem was more prominent in small and medium span bridges due to the large proportion of the vehicle live loads in the total load effect.”, which had been revised to Section 1.
3. What are slab beams? is this a technical term in the Chinese standards as it does not appear to be common technical language and hence is confusing without further description.
Reply: Reinforced concrete plate-girders are commonly used in medium and small span Bridges in China, usually it is a solid beam with rectangular section, as is shown in Fig.5 in our manuscript. So, “slab beams” had been revised as “plate-girders”, please see our revisions.

Reviewer 2 Report
The article entitled “The Mechanical Properties of Reinforced Concrete Slab Beams when Placed Under Repeated Simulated Vehicle Loads” investigated the mechanical performance of reinforced concrete slab beams when subjected to long-term cyclic vehicle loading. The mechanical behavior of reinforced concrete slab beams under cyclic loading proved to be largely determined by the rebar. The peak value of the cyclic loads had a strong influence on the failure state and performance degradation law of the beams. With an increase in overload ratio, the mid-span deflection and residual deflection increased gradually, and the bending stiffness of the beam decreased significantly, while the cumulative damage increased gradually. During the entire cyclic loading process, the tensile strain of the rebar and the compressive strain of the beams’ concrete was less than their ultimate strain, and the ultimate failure of the beam was caused by the sudden failure generated by the fatigue fracture of the tensile reinforcing bar. The overload ratio of vehicle loads should be strictly controlled in order to avoid or reduce the possibility of highway bridges experiencing structural brittle failure.
The paper is interesting.
However, the Abstract of the paper is too long and could be better written.
Author Response
1. The Abstract of the paper is too long and could be better written.
Reply: The Abstract in the manuscript had been written in a more technical manner, please see our revisions on p. 1.
“The effect of vehicle loads on reinforced concrete plate-girders was evaluated using the current Chinese specifications. Repeated loading performance tests with loading amplitudes of 77kN, 97kN, and 121kN, which corresponding to the standard vehicle load, 1.25 times overload and 1.6 times overload proportions effect were carried out on three full-scale simply-supported reinforced concrete plate-girders. Our research results indicate that the development of cracks in reinforced concrete beams can be divided into three stages: rapid development, stability and failure. During the entire process, the strain of steel and concrete did not reach their yield strain. The most severe damage done to the concrete beams was the brittle fractures caused by the fatigue fracturing of the rebar. When in a stable condition, the extent to which the vehicle was overloaded had a significant effect on the fatigue performance of the beam, and the corresponding residual deflection and residual strain increased with the rise in the overload proportion. In addition, as the overload proportion increase, the stiffness degradation and the cumulative damage that occurred under the same loading cycle is more significant. The test beam reached failure after being subjected to 350,000 and 670,000 repeated loading cycles, when the load was 1.6 times and 1.25 times of the standard load effect. With a standard vehicle load effect, the test beam was able to endure 2,000,000 repeated load cycles with no significant degradation in stiffness and bearing capacity.”

Round 2
Reviewer 1 Report
The manuscript can be accepted in terms of technical content but should undergo further English language editing.
Reviewer 2 Report
The paper can now be accepted.